# Short-Term Ocean Rise Effects on Shallow Groundwater in Coastal Areas: A Case Study in Juelsminde, Denmark

Ronja Forchhammer Mathiasen *, Emilie Padkær Haugan, Theis Raaschou Andersen, Henriette Højmark Hansen, Anna Bondo Medhus and Søren Erbs Poulsen

Research Centre for Built Environment, Energy, Water and Climate, VIA University College, 8700 Horsens, Denmark; ehau@via.dk (E.P.H.); thra@via.dk (T.R.A.); henriette@jokiha.dk (H.H.H.); aeu@energinet.dk (A.B.M.); soeb@via.dk (S.E.P.)
* Correspondence: rocf@via.dk; Tel.: +45-87554203

**Abstract:** Coastal areas situated at lower elevations are becoming more vulnerable to flooding as a result of the accelerating rise in the global sea level. As the sea level rises, so does the groundwater. Barriers designed to shield against marine flooding do not provide protection against flooding caused by rising groundwater. Despite the increasing threat of groundwater flooding, there is limited knowledge about the relationship between sea level rise and groundwater fluctuations. This hinders the ability to adequately consider sea level rise-induced groundwater flooding in adaptation initiatives. This study aims to investigate how local groundwater in Juelsminde, Denmark, responds to changes in sea level and to evaluate the predictability of these changes using a machine learning model. The influence of the sea on the shallow groundwater level was investigated using six groundwater loggers located between 45 and 210 m from the coast. An initial manual analysis of the data revealed a systematic delay in the rise of water levels from the coast to inland areas, with a delay of approximately 15–17 h per 50 m of distance. Subsequently, a support vector regression model was used to predict the groundwater level 24 h into the future. This study shows how the groundwater level in Juelsminde is affected by sea level fluctuations. The results suggest a need for increased emphasis on this topic.

**Keywords:** global sea level rise; groundwater fluctuations; machine learning model; predictability; loggers





## 1. Introduction

The global sea level rise is accelerating due to the melting of ice sheets and the thermal expansion of the oceans as a result of climate change [1–3]. This will affect millions of people living in low-lying coastal areas worldwide [1,4,5], and many urban coastal areas are already experiencing the consequences of increased precipitation, such as more frequent storm events and sea level rise (SLR) [6].

While SLR poses direct challenges to coastal areas, such as flooding and erosion [7–9], groundwater inundation (GWI) is an indirect and increasingly problematic consequence of SLR [8]. The groundwater level (GWL), especially in permeable ground, responds to tidal forces. This narrowing of the unsaturated space between the GWL and infrastructure may lead to GWI. Extremely high tides and SLR will intensify these floods due to a higher groundwater table and reduced unsaturated space to store water from storm events [6,10–13]. There may be many contributors that can influence GWI, such as rainfall, temperature, geology, and high tides during a storm event. Furthermore, low-lying coastal zones are prone to compound flooding, where marine and/or surface flooding can happen simultaneously with GWI [4,9]. Coastal barriers designed to protect against surface water are not effective in preventing flooding from below, making GWI especially problematic. This leaves buildings, basements, and both underground and surface infrastructure at risk of flooding [6,8,11,12,14,15]. Consequently, it can lead to contamination of the surface and groundwater with sewage [10].

The role of groundwater is important to fully understand the effect of SLR in coastal areas; however, it is often inadequately addressed in studies related to this topic. The effects of SLR are often assessed using numerical modelling, but there are few studies on the modelling and forecasting of GWI. This lack of information on GWI in coastal areas means that it is not being adequately considered in planning or adaptation initiatives [6,8,9,12,16]. Modelling GWI requires consideration of all parameters that may influence the GWL during extreme weather events, as well as the risk of compound flooding [9]. Thus far, many of the models used to investigate these systems have been based on physical principles. However, these models require substantial amounts of information about the system, and even when the data are available, they can be difficult to calibrate. Therefore, machine learning models are gaining popularity among hydrologists as screening tools or supplements to hydrological models, as they perform well and require less data input compared to the results they provide [6].

Existing research papers on the effects of SLR on GWL portray a variety of methods and perspectives, reflecting the complexity of the topic.

Habel et al. [10] used a groundwater flow model to assess the impact of SLR and high tides on GWI in Waikiki, Honolulu, Hawaii and concluded that an SLR of only approximately 1 m would result in a significant increase in the surface area experiencing GWI and its corresponding ramifications. Bjerklie et al. [12] also found that SLR causes groundwater rise (GWR). Using a 3D groundwater flow model, researchers discovered that in New Haven, Connecticut, a 0.91 m SLR scenario resulted in a corresponding 0.91 m increase in the near-coast GWL. Furthermore, the model indicated that even a GWL ranging from 5.2 to 7.3 m above sea level responded to the SLR. Results from Knott et al. [15] using a groundwater flow model showed that in coastal New Hampshire, USA, there is an estimated mean rise in GWL of 66% of SLR between 0 and 1 km from the coastline. Additionally, the study found that there is a response in GWL up to 5 km inland (3% of SLR) from the coastline. Similar among these three studies is the conclusion that the models must include more parameters and/or data to enhance their reliability [10,12,15].

In Bowes et al.'s [6] study, two machine learning models, namely, long-short-term memory (LSTM) and recurrent neural networks (RNNs), were investigated for their effectiveness in forecasting and modelling GWL in Norfolk, Virginia. A comparison of the models showed that LTSM networks are useful in real-time operational forecasts, and authors predict that GWL forecasting will become a valuable tool in the future management and modelling of coastal flooding.

Although the described studies have addressed this topic in different localities around the world, no systematic research has been conducted on the impact of SLR on GWL in Denmark.

Therefore, this study aimed to investigate how the local groundwater table responds to fluctuations in sea level and to uncover regional variations in the dependence on sea level and precipitation in the town Juelsminde in Denmark. Finally, the predictability of groundwater fluctuations was tested using a simple machine learning model based on the information mentioned above.

## 2. Materials and Methods

### 2.1. Study Area and Geological Setting

Juelsminde is located on the eastern coast of Jutland, Denmark, on a peninsula bordered by the Kattegat Sea to the east and Vejle Fjord to the south (Figure 1). Juelsminde is prone to flooding and has been identified as one of the top 10 cities in Denmark that are most at risk of flooding [17]. The nearshore areas are dominated by vacation homes, while the harbour is located in the northern part of the city. Further inland, residential houses dominate. A dike with an elevation of 2.6 m above sea level (masl) protects the buildings located south of the harbour from flooding caused by storm surges. The groundwater level is only 0–1 m below the ground surface (mbgs), and large parts of the city are drained through a series of ditches, from which water is pumped over the dike into the Kattegat Sea.

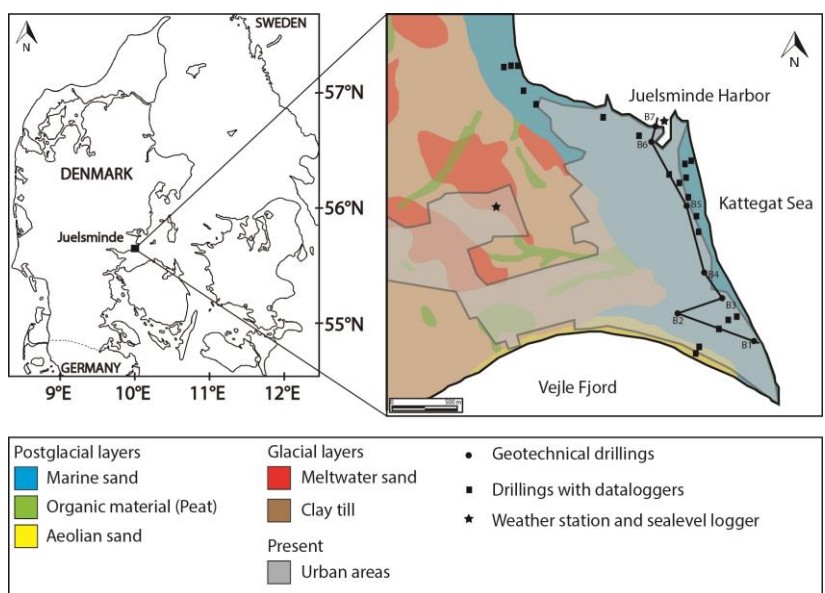

**Figure 1.** Overview map showing the Juelsminde area, the soil types, the position of the wells and geotechnical drillings in Figure 2, and the location of the dataloggers.

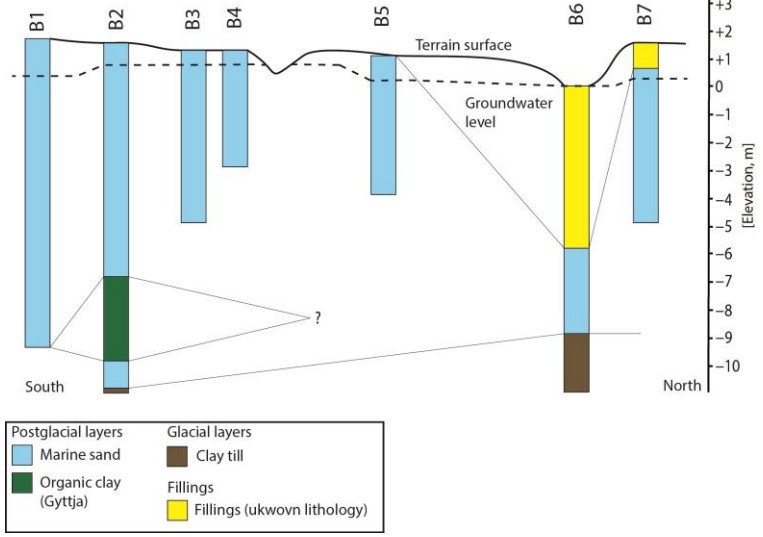

**Figure 2.** Cross-section through Juelsminde showing borehole lithologies.

In Figure 1, the near-surface soil types are shown. The study area is characterised by glacial sediments in an elevated area to the west, covered by postglacial sediments in a lower lying area to the east. Figure 2 displays a geological cross-section based on well data. The glacial till observed in drilling B2 and located towards the west in Figure 1 was deposited during the Pleistocene glaciations, when the area was repeatedly covered by ice sheets [18]. The postglacial sediments are dominated by marine sand and, to a lesser extent, organic clay (Gyttja). The marine sand was deposited during the Holocene period, when ice melted and retreated, following the latest glaciation [19]. The sea level rose, causing low-lying areas in Denmark to flood based on the combined effect of the sea level rise and the land still being depressed from the weight of the ice sheets. The land has a much slower response to reaching equilibrium than the ocean. The flooding peaked during the Littorina transgression, which occurred approximately 8000 years ago [20]. This event is responsible for the lower marine sand layer observed in drilling B2. Marine conditions were established and the gyttja layer found in drilling B2 was deposited. The thickness of the lower marine sand is approximately 1 m, but it may be greater in areas where no

organic clay is present. The organic clay layer has a thickness of 3 m and is composed of clay (Gyttja) containing plant remains and shells.

As the land rose following the ice retreat, the shoreline shifted towards its current position. During this regression, the upper layer of marine sand was deposited. It is up to 10 m thick and described as fine- to medium-grained, well-sorted sand containing shells.

The anthropogenic fillings in the area, typically 1–4 m thick, are expected to be local occurrences related to constructions and, as expected, the greatest thickness is found in the harbour area.

*2.2. Data*

Since 2017, six Rotek loggers have been measuring the GWL every 10 min, in addition to the manual measurements carried out every second week to validate the data. They are located along coast-perpendicular profiles, with distances from the coastline between 45 and 210 m (Figure 3).

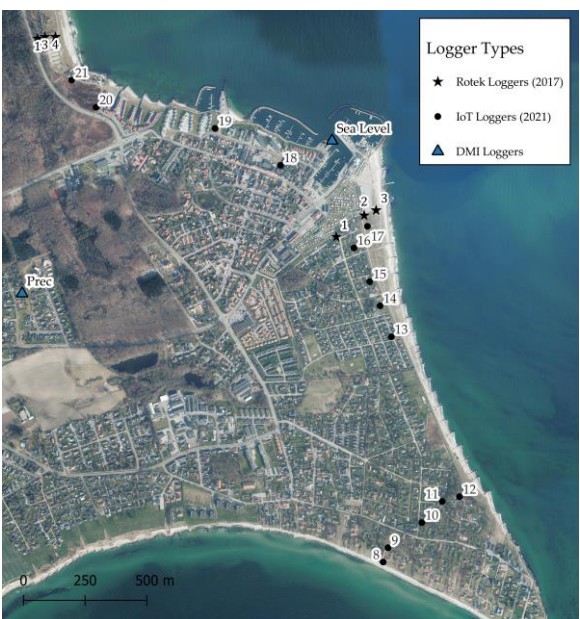

**Figure 3.** Location of the different logger types installed in the Juelsminde area.

The northernmost profile is located in an open area devoid of significant buildings, whereas the southern profile extends from the beach just south of the harbour and into the urbanised region behind the dike. In the last quarter of 2021, 14 additional IoT groundwater loggers were installed to investigate the dynamics along the coastline and gain a more comprehensive understanding of spatial variation. Tide gauges were installed along the coastline and in residential areas near the coast, as shown in Figure 3. The Danish Meteorological Institute (DMI) conducts sub-hourly measurements of the sea level in Juelsminde harbour. The precipitation amounts in Juelsminde are also recorded by the DMI. The data are freely available.

*2.3. Data Preparation*

The various sources of data require different levels of data processing, as shown in Figure 4. The data obtained from the six Rotek loggers were all corrected for barometric pressure. Where only a few values were missing, they were calculated using linear interpolation. Major outliers were removed (<0.02% of the dataset), and staggered sections were aligned with the rest of the data series. The sources of sudden jumps (offsets of sections) in the data are unknown, which means that there is a risk of correcting the wrong section of the data. This uncertainty is one of the reasons why this study focused on the relative

changes in the groundwater table. The mean of the data series was subtracted, allowing easier comparison between the different groundwater time series. The data retrieved from the 14 IoT loggers were calibrated and referenced to the appropriate elevation below sea level (DVR90) [21], and significant outliers were removed by the company that, in addition, provided the groundwater loggers. The sea level and precipitation data from the DMI required very little correction. Only a few outliers were removed. All data series that were compared were first standardised to the same resolution before analysis, using either linear interpolation or downscaling.

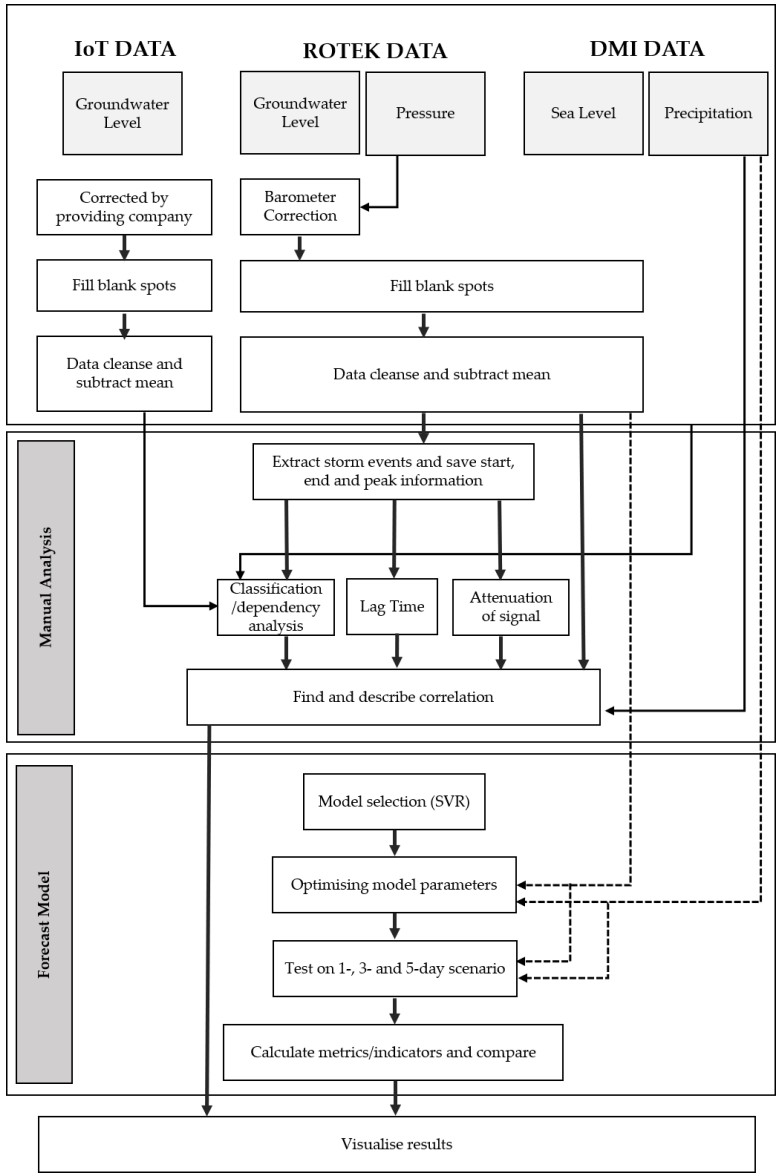

**Figure 4.** Workflow of the data treatment, analysis, and model process. The dashed arrows representing the data used for the forecast model.

In this study, two different analyses were carried out: (1) A manual analysis was conducted to examine the groundwater responses to high tide and storm events separately. Based on the results of the manual analysis, (2) a machine learning model was constructed to assess the predictability of the groundwater response.

Manual Analysis of the Sea Level and Groundwater Interaction

Rises in shallow groundwater levels resulting from large, sudden changes in sea level are typically more distinct and easier to identify and separate from each other than minor variations caused by tidal fluctuations. Additionally, they can be traced further inland, making them ideal for evaluating the systemic response of shallow groundwater to sea level rises. The most significant sea level rises were identified and extracted for analyses.

The analysis was conducted on the six data loggers (Rotek), with the longest available time series, placed in two profiles perpendicular to the coast. The sea level rise was determined by subtracting the mean sea level from the level recorded over the past two days. Through automated peak identification, time periods with a peak prominence over 1.2 m and an SL above 0.5 m were selected and are listed in Table 1. Information regarding the scale of the SLR and the time interval covering the period of five days before the peak sea level and the subsequent seven days was gathered. This selection resulted in 30 high-tide events with relative amplitudes ranging from 1.22 to 1.96 m. An example of the automatic selection of high-sea-level events is shown in Figure 5.

**Table 1.** An overview of the data sources used in this study.

| Logger Type | Logger No. | Parameter | LON | LAT | Distance to Coast (m) | Start Time |
|---|---|---|---|---|---|---|
| Rotek loggers | 1 | Barometer | 55.719466 | 9.997424 | 139 | 27 September 2017 |
| | 2 | GWL | 55.719519 | 9.997907 | 139 | 1 August 2018 |
| | 3 | GWL | 55.719519 | 9.998589 | 101 | 27 September 2017 |
| | 4 | GWL | 55.712096 | 10.01651 | 52,5 | 27 September 2017 |
| | 5 | GWL | 55.712857 | 10.018334 | 210 | 27 September 2017 |
| | 6 | GWL | 55.713039 | 10.019108 | 91 | 27 September 2017 |
| | 7 | GWL | 55.715592 | 10.016333 | 45,5 | 27 September 2017 |
| IoT loggers | 8 | GWL | 55.700241 | 10.019238 | 25 | 9 November 2021 |
| | 9 | GWL | 55.700759 | 10.019567 | 85 | 9 November 2021 |
| | 10 | GWL | 55.701656 | 10.021771 | 237 | 9 November 2021 |
| | 11 | GWL | 55.702416 | 10.023098 | 166 | 9 November 2021 |
| | 12 | GWL | 55.702576 | 10.024219 | 95 | 9 November 2021 |
| | 13 | GWL | 55.708428 | 10.019977 | 93 | 9 November 2021 |
| | 14 | GWL | 55.70956 | 10.019282 | 93 | 7 December 2021 |
| | 15 | GWL | 55.710447 | 10.018632 | 111 | 7 December 2021 |
| | 16 | GWL | 55.711681 | 10.01765 | 153 | 7 December 2021 |
| | 17 | GWL | 55.712459 | 10.018539 | 90 | 10 November 2021 |
| | 18 | GWL | 55.714725 | 10.012991 | 106 | 10 November 2021 |
| | 19 | GWL | 55.716101 | 10.008804 | 75 | 9 November 2021 |
| | 20 | GWL | 55.716932 | 9.999464 | 122 | 7 December 2021 |
| | 21 | GWL | 55.717932 | 10.001148 | 67 | 7 December 2021 |
| DMI loggers | 100 | Sea level | 55.7156 | 10.0163 | - | 27 September 2017 |
| | 101 | Precipitation | 55.7102 | 9.9962 | - | 27 September 2017 |

A minimum threshold of 1.2 m for relative water level change ensures a high enough signal-to-noise ratio inland while also including a sufficient number of high-tide events to evaluate possible correlations. The process of peak selection and the subsequent analysis methods were inspired by Bowes et al. [6]. Modified versions of the available code were used. The resulting peak events are presented in Table 2.

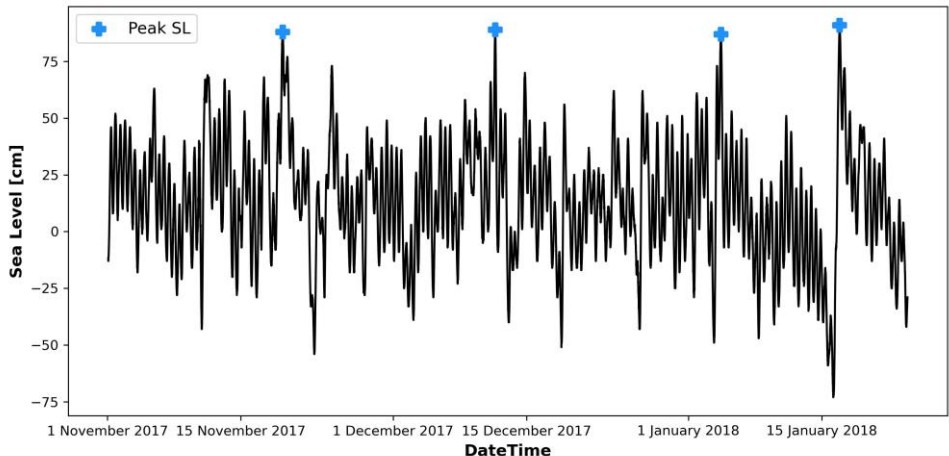

**Figure 5.** An example of the automatic selection of high-sea-level events. The blue crosses represent peak levels exceeding the minimum criteria.

**Table 2.** The selected SLR events and the corresponding peak values.

| Event No. | Start Datetime | Peak Datetime | End Datetime | Peak Prominence [cm] | SL at Peak (cm) (DVR90) |
|---|---|---|---|---|---|
| 1 | 7 October 2017 | 13 October 2017 | 19 October 2017 | 131 | 79 |
| 2 | 13 October 2017 | 18 October 2017 | 25 October 2017 | 128 | 76 |
| 3 | 14 November 2017 | 19 November 2017 | 26 November 2017 | 131 | 88 |
| 4 | 6 December 2017 | 11 December 2017 | 18 December 2017 | 143 | 89 |
| 5 | 30 December 2017 | 4 January 2018 | 11 January 2018 | 138 | 87 |
| 6 | 11 January 2018 | 16 January 2018 | 23 January 2018 | 164 | 91 |
| 7 | 23 January 2018 | 28 January 2018 | 4 February 2018 | 135 | 74 |
| 8 | 7 February 2018 | 12 February 2018 | 19 February 2018 | 166 | 68 |
| 9 | 11 February 2018 | 16 February 2018 | 23 February 2018 | 142 | 60 |
| 10 | 22 February 2018 | 27 February 2018 | 6 March 2018 | 142 | 60 |
| 11 | 11 March 2018 | 16 March 2018 | 23 March 2018 | 142 | 60 |
| 12 | 13 November 2018 | 18 November 2018 | 25 November 2018 | 134 | 55 |
| 13 | 29 November 2018 | 4 December 2018 | 11 December 2018 | 175 | 109 |
| 14 | 3 January 2019 | 8 January 2019 | 15 January 2019 | 165 | 136 |
| 15 | 13 January 2019 | 18 January 2019 | 25 January 2019 | 124 | 74 |
| 16 | 6 February 2019 | 11 February 2019 | 18 February 2019 | 148 | 79 |
| 17 | 3 March 2019 | 8 March 2019 | 15 March 2019 | 162 | 104 |
| 18 | 11 March 2019 | 16 March 2019 | 23 March 2019 | 123 | 71 |
| 19 | 13 March 2019 | 18 March 2019 | 25 March 2019 | 134 | 76 |
| 20 | 10 September 2019 | 15 September 2019 | 22 September 2019 | 156 | 108 |
| 21 | 23 October 2019 | 28 October 2019 | 4 November 2019 | 123 | 79 |
| 22 | 24 November 2019 | 29 November 2019 | 6 December 2019 | 164 | 111 |
| 23 | 2 December 2019 | 7 December 2019 | 14 December 2019 | 122 | 102 |
| 24 | 7 December 2019 | 12 December 2019 | 19 December 2019 | 136 | 86 |
| 25 | 30 January 2020 | 4 February 2020 | 11 February 2020 | 139 | 93 |
| 26 | 4 February 2020 | 10 February 2020 | 16 February 2020 | 196 | 136 |
| 27 | 18 February 2020 | 23 February 2020 | 1 March 2020 | 149 | 102 |
| 28 | 8 March 2020 | 14 March 2020 | 20 March 2020 | 124 | 79 |
| 29 | 24 March 2020 | 29 March 2020 | 5 April 2020 | 124 | 75 |
| 30 | 11 September 2020 | 17 September 2020 | 23 September 2020 | 126 | 78 |

*2.4. Response Time from the Sea Level Rise*

To determine the lag time between the rise of the sea level and groundwater, an automated algorithm was used to calculate the time from the peak of the sea level rise to the peak of the groundwater level rise. Each storm event was analysed by extracting the time interval from both sea level and groundwater level data, with the sea level data being

interpolated to match the higher resolution of the groundwater data. The raw data were compared to two filtered datasets using two different methods: (1) A Butterworth low-pass filter and (2) a running mean. These methods were employed to eliminate high frequencies, such as tidal changes, from the data. However, filtering affects the peak amplitude and, therefore, was only used to determine the signal delay. Applying a running mean proved to perform best on this dataset. The delay was found using cross-correlation of single loggers with the sea level data. All of the results were verified manually.

### 2.5. Attenuation of Groundwater Peak

The inland amplitude damping of the sea level rise indicates the extent to which a storm surge of a certain size affects the groundwater table. This attenuation was calculated for each event by determining the relative increase in the groundwater level at each logger location. The relative increase in GWL was calculated by subtracting the average GWL measured from 48 to 24 h prior to the GWL peak from the GWL at peak time.

### 2.6. Predictive Model

A delayed signal from the SLR and corresponding GWR enables the prediction of the GWL based on the sea level. The approach involved using a quick and simple model to gain insight into its predictive capabilities.

The primary objective of developing a predictive model is to estimate future increases in the groundwater level by considering current and forecasted changes in the sea level. The aim was to construct a model that could provide precise forecasts, and thus, the focus was primarily on calculating changes in the groundwater level one day in advance.

Based on initial tests using the same input in several simple machine learning models, support vector regression (SVR) provided the most accurate results. An SVR model is a regression model that determines the best fit based on a defined, acceptable error [22]. The model presented in this article is based on a linear kernel.

The model was partly based on historical data from the DMI and the monitored groundwater level data, with sea level and groundwater observations used as input. The DMI creates weather and sea level forecasts up to five days in advance. These forecasts can be utilised to generate predictions for the groundwater level, and based on these predictions, a warning system can be developed for areas at risk.

The model was trained on data from each of the six loggers individually, and the results were calculated separately for each logger. Hence, in this study, the model functioned as six separate models. The model's predictive capabilities include 24 h, three-day, and five-day forecasts. The input parameters were selected in part based on the initial manual analysis of the data and the response of the groundwater.

In the final model, the input parameters were as follows:

- Groundwater level from the past four days;
- Sea level from the past four days and one day into the future (using the DMI's one-day forecast);
- Precipitation from the past 24 h and one day into the future (using the DMI's one-day forecast);
- Sine and cosine curve representing seasonality.

The model was trained on 60% of the available data, while the validation and cross-validation data each consisted of 20%.

The indicators' adjusted $R^2$ ($R^2_{adj}$) and predicted $R^2$ ($R^2_{pred}$) were used to compare the model performance. $R^2_{adj}$ indicates how well the terms fit a curve or line, but it adjusts for the number of parameters in a model. This means that if a parameter is included in the model but does not contribute to the model, the $R^2_{adj}$ will decrease. If the parameter

contributes to the model, $R^2_{adj}$ will increase. $R^2_{adj}$ will always be less than or equal to $R^2$. $R^2$ is the residual sum of squared errors divided by the total sum of squared errors.

$$R^2 = 1 - \frac{SS_{RES}}{SS_{TOT}} = 1 - \frac{\sum_i (y_i - \hat{y}_i)^2}{\sum_i (y_i - \overline{y}_i)^2} \qquad (1)$$

where $y_i$ is the dependent variable for observation $i$.

$$R^2_{adj} = 1 - \frac{(1 - R^2)(n - 1)}{n - k - 1} \qquad (2)$$

where $n$ is the sample size and $k$ is the number of variables in the model.

$R^2_{pred}$ is also known as the PRESS statistic and is used to determine how well a regression model makes predictions. If the data consist of a lot of noise, then $R^2_{pred}$ will be low, as it is not possible to predict random noise. $R^2_{pred}$ is useful to avoid overfitting.

$$R^2_{pred} = 1 - \left( \frac{PRESS}{SS_{TOT}} \right) = 1 - \frac{\sum_i \left( y_i - \hat{y}_{(-i)} \right)^2}{\sum_i (y_i - \overline{y}_i)^2} \qquad (3)$$

where $\hat{y}_{(-i)}$ is the predicted value of the response variable for this observation found from the fitted regression equation [23,24].

## 3. Results and Discussion

### 3.1. Groundwater Wave Characteristics

In a homogeneous medium, we expect to find a linear relationship between distance and time for the propagation of a wave peak. Figure 6 displays a boxplot representing the observed delay times at the loggers.

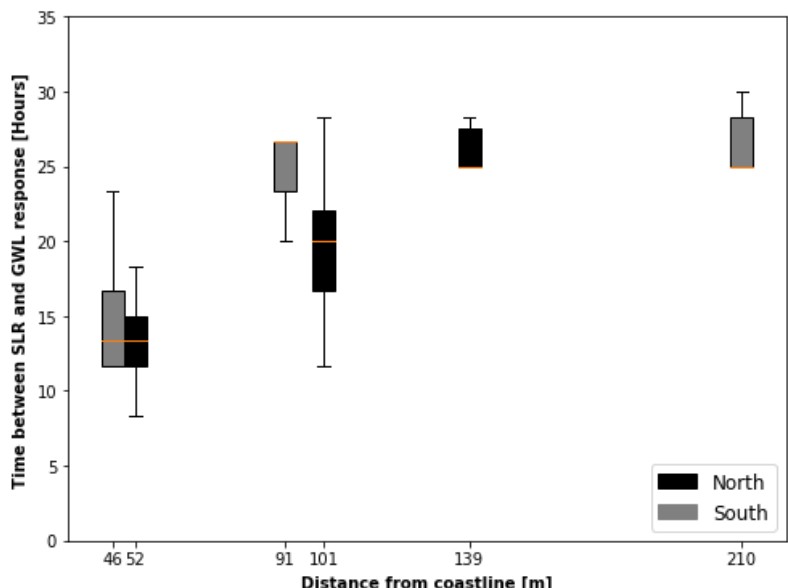

**Figure 6.** Boxplot (with medians marked in orange and 25% and 75% quartiles) visualising the lag times for the delayed responses in the groundwater table initiated by a sudden rise in the sea level in the northern (black) and the southern (grey) profiles.

The loggers along the northern area, which are not affected by dikes or buildings, appeared to exhibit a linear relationship with a signal delay of approximately 14 h and 50 m to the coastline, reaching logger 2, 139 m from the coast, 25–26 h after the SLR. The two most coastal positions on the southern profile also appeared to reasonably follow the

trend, while the farthest logger (logger 5) seemed to be influenced by other factors, with a delay time ranging between 25 and 30 h. This can be explained by the natural, high variation in the time lag response of the GWL, as the lag time is expected to be dependent on the magnitude of the sea level rise. The peak response in the GWL was clearly visible in the coastal loggers, but the algorithm had difficulty identifying the resulting rise in the loggers located further away from the coast. This is due to the low signal-to-noise ratio in the logger data, which made it difficult to identify. Moreover, the further inland the loggers are located, the greater the possibility that the signal from the SLR is drowned out by the responses from precipitation events. Furthermore, the further inland, the more likely that local pumping and drainage also affect the local groundwater level. It is worth noting that logger 5 is the only one placed on the inner side of the dike and within a constructed area. This may involve the presence of various low-conductivity barriers, which affect both the attenuation of the wave amplitude and delay the signal response in the groundwater well [25]. Consequently, this may account for the difficulty in detecting the signal in logger 5. Additionally, Sánchez-Úbeda et al. [26] suggested that the depth of the diver within the well can influence both the amplitude and the delay of the signal. In lower sections of the aquifer, the signal would experience less damping and travel at a faster pace. This observation is supported by previous studies [27,28]. In this study, the divers were located in the shallow aquifer, ranging from 121.4 cm (logger 6) to 297.8 cm (logger 5) in depth, with an average of 228 cm. Although the depth can potentially affect measurements, it is likely that this effect is more pronounced in deeper aquifers with higher pressure. Therefore, it is plausible that the low-conductivity layer associated with the dike or local infrastructure influenced the signal in well 5.

Figure 7 shows that the amplitude of the groundwater level decreased as the distance from the coast increased. We expect the amplitude damping in a free aquifer to follow a logarithmic function [29]. With a logarithmic time axis, the relationship could be estimated by fitting a straight line. However, there were outliers, especially when very large sea level rises occurred, resulting in extra-high amplitudes. Note that events 14 and 25 had outliers caused by possible mispicks. However, there were no sufficient arguments to exclude them from the dataset. A number of events (2, 7, 15, 18, and 21) exhibited no noteworthy increase in the GWL that could be attributed to the SLR. This may be due to a limited potential for a rise in the sea level. If an aquifer is limited by the topography, a groundwater table close to the terrain will leave little room to store more water and hence will not have the capacity to let the groundwater table rise in response to the sea level [30,31]. All of the events occurred in either October, January, or early March, which tend to be cold and wet months in Denmark [32]. Among these events, three (2, 18, and 19) experienced significantly above-average precipitation in either the previous or the current month, with September 2017 witnessing a 51% excess rainfall compared to the 1991–2020 average in Juelsminde, followed by a 128% deviation in March 2019 and a 62% deviation in October 2019 [33]. These excessive precipitation events may have caused the shallow groundwater to be so close to the terrain that it reached its topographic limitation. However, event 17, occurring just a week before event 18, produced a response in the groundwater loggers, despite similar topographic limitations. Event 17 exhibited a substantially higher SLR (162 cm) compared to event 18 (123 cm), ranking as the seventh highest and third lowest, respectively, among the listed events. Furthermore, the remaining four events without a response also fell within the lowest quarter (except event 7) when ordered by sea level rise (as depicted in Figure 8). This suggests that the currently defined lower threshold for significant sea level rises may be set too low, leading to the absence of a response in the groundwater level during these events.

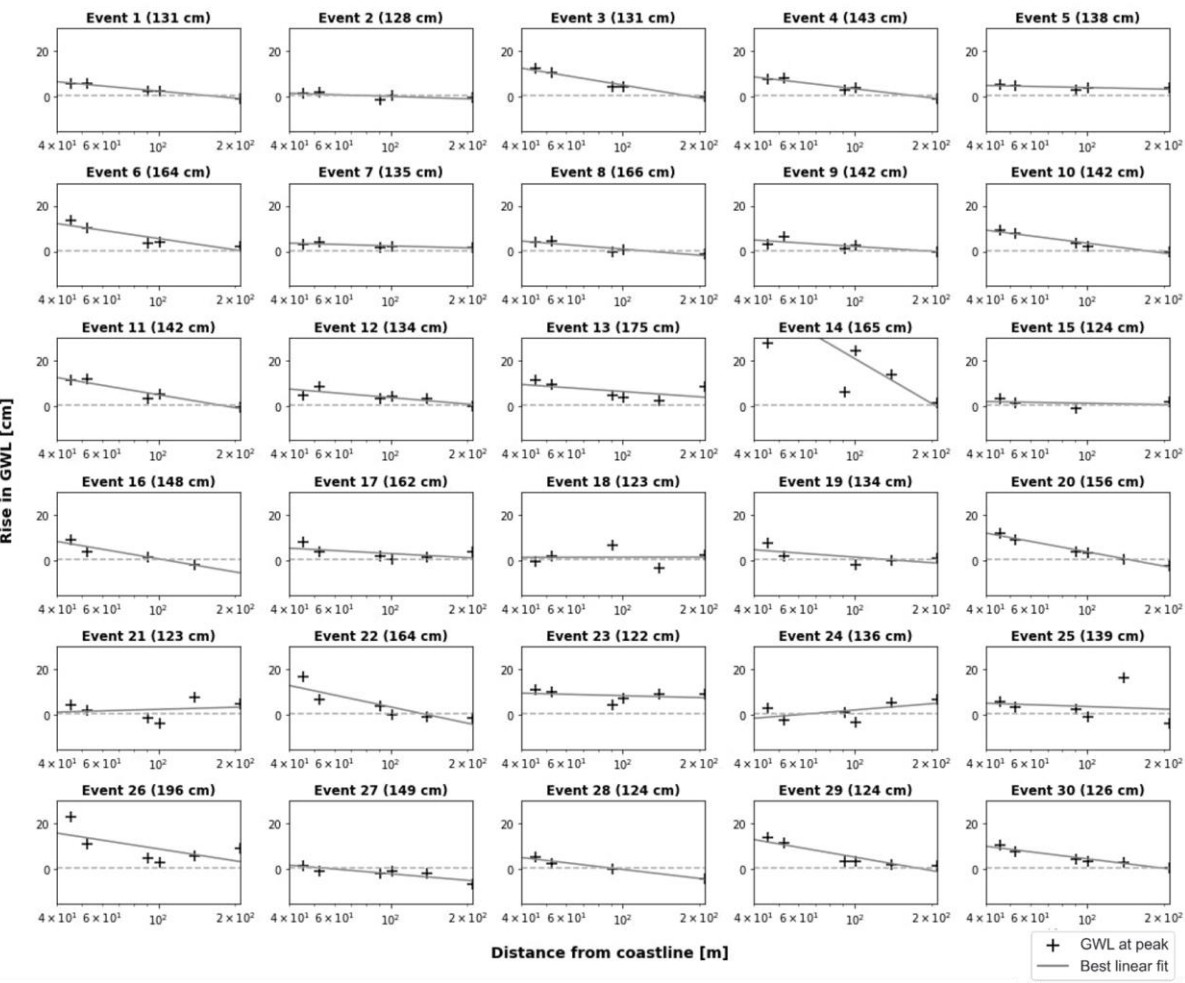

**Figure 7.** The relative rise in the ground-water level monitored in the wells for each of the 30 SLR events.

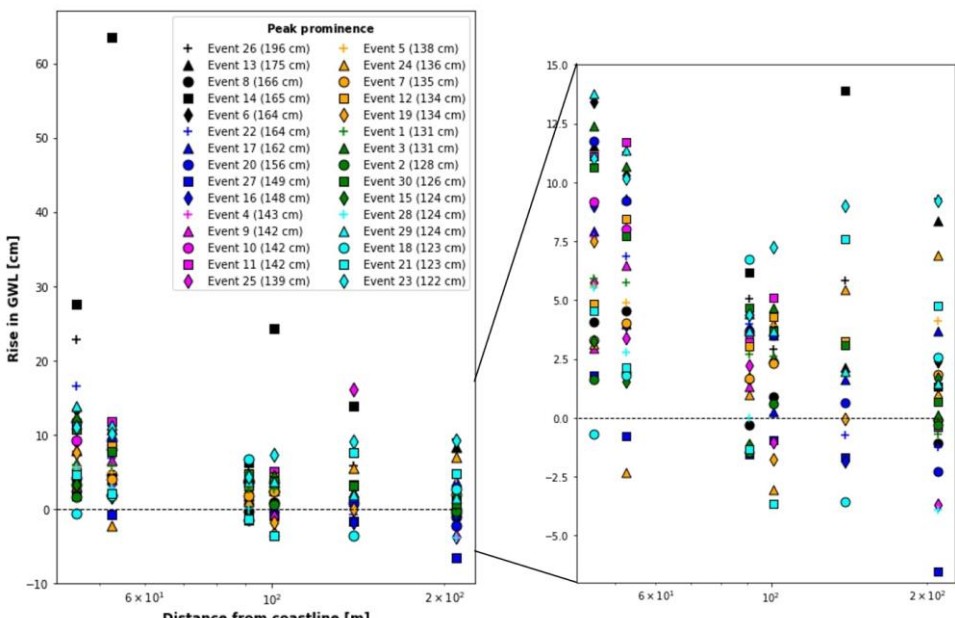

**Figure 8.** The relative rise in groundwater level registered in the wells during 30 SLR events sorted by the peak prominence of the SLR event.

Figure 8 displays the individual measurements from each event, sorted by the size of the SLR. The largest events generally showed the strongest responses in the GWL, but otherwise, there was no significant pattern amongst the smaller events. However, based on the observation made from Figure 7, eliminating the bottom 10 events may result in a cleaner dataset, revealing stronger correlations.

Figure 9 indicates a semi-linear relationship between the rise in sea level and the corresponding increase in the GWL. This relationship was particularly evident in the most prominent SLR events and in logger 7, which is closest to the coast. Although the relationship was also detectable in loggers 3 and 6, it was generally significantly less well defined in the two farthest loggers due to signal reduction. However, these trends are not entirely straightforward. There was significant variation among different events in terms of both the determination of phase shift and the change in amplitude. This is due to the simplification of the groundwater system in Juelsminde and the expectation of complete reliance on the ocean, particularly as the distance from the coast increases. The farther from the coast, the greater the influence of additional factors.

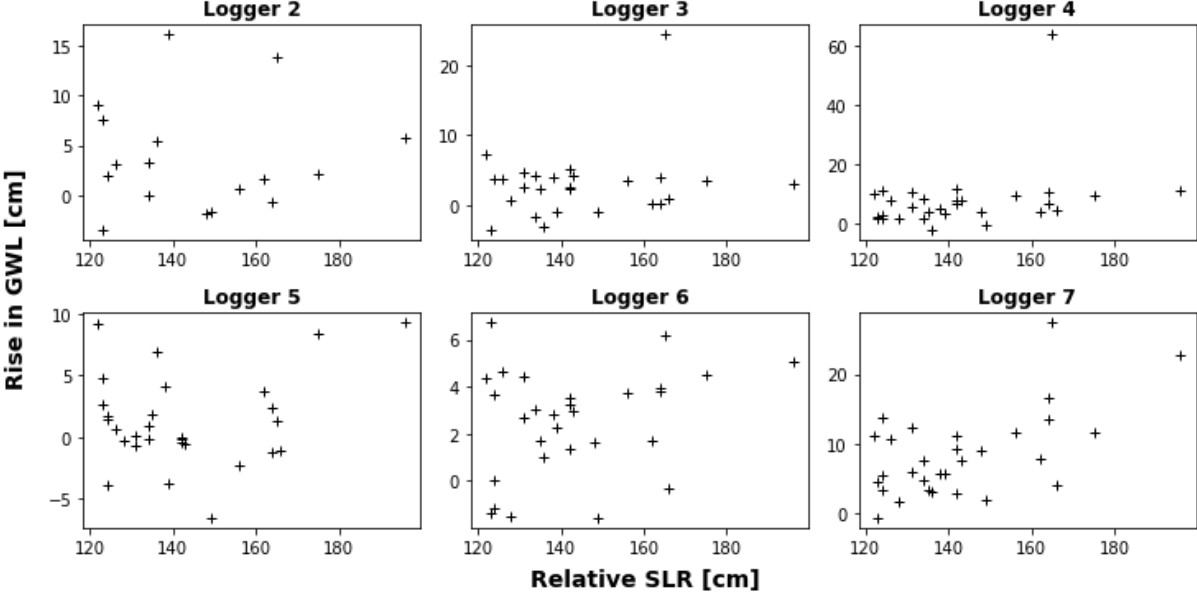

**Figure 9.** The relative rise in the GWL plotted against the SLR peak prominence for all SLR events in the six wells from 2017.

### 3.2. Sensitivity and Dependency Analysis

The 14 new loggers did not have time series long enough to contain sufficient events for the previous analysis. Instead, the data were used to conduct an overview analysis of the groundwater's dependency on and sensitivity to sea level rises and precipitation events (Figure 10). The dependence on sea level fluctuations is determined not only by the storm surge responses, but also by the trace of tidal changes in the loggers. In these cases, precipitation has a reinforcing effect on the signal, but it is not the primary cause (blue circles). When the GWL cannot be correlated with sea level changes but mainly responds to precipitation, it belongs in the red (triangle) category. Half of the wells seemed to be equally influenced by the two parameters (black squares) and one logger provided too little data to categorise the well (white circle). In general, those loggers located closer to the coast tended to be more vulnerable to the impact of sea level changes compared to those situated further inland. In certain areas, this principle may not apply to individual loggers, and it is presumed that local conditions may influence which parameters have the greatest impact on the groundwater level in a logger. Hence, it may be difficult to determine exactly how much different parameters affect the various wells, but it can be generally stated that precipitation affects the GWL in all wells to varying degrees, while the impact of sea level

is highly dependent on the distance to the coast. The sea level's dependence is also an indication of the hydraulic connection between the well location and the ocean. Well no. 20 is located relatively far inland, but it clearly responds to tidal changes. It is located adjacent to a wetland that is connected to the ocean, which explains the rapid response. Well 14 appears to be primarily influenced by precipitation, despite no significant distinctions in its location compared to the neighbouring loggers 13 and 15. The cause of this discrepancy could be caused by either geological factors or local infrastructure. It is possible that there is an obstruction along the pathway from the coast to well 14, hindering the hydraulic connection. Alternatively, the contribution from precipitation in that specific area may be notably higher, leading to a greater accumulation of surface water. The abrupt transition from the blue to orange categories between loggers 8 and 9, as well as between 12 and 11, could be a result of topographical variations or the presence of infrastructure. Logger 8, located in close proximity to the shoreline and at a low elevation, might be affected by both topography and potential obstructions caused by infrastructure. Logger 9, situated within a built-up area, could experience a blockage in its hydraulic connection to the ocean or gather more surface water during precipitation events due to local construction. Logger 12 is positioned in an area with few buildings and no roads obstructing the path to the ocean. On the contrary, logger 11, located nearby, is surrounded by houses, which seemingly has a considerable impact. Further investigation into the types of infrastructure that have the greatest influence on sensitivity to precipitation and sea level rise would be of interest.

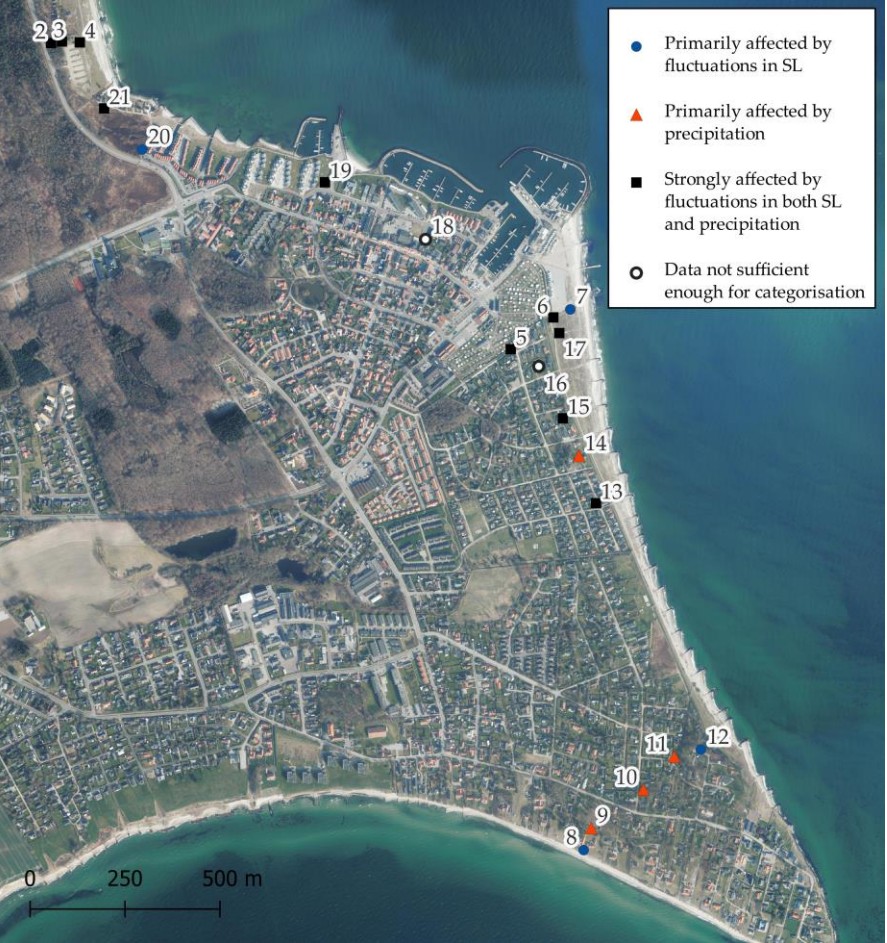

**Figure 10.** The classification results for the GWL in each well depending on the SLR or precipitation event.

### 3.3. Predictability of Groundwater Fluctuations

Results from the 24 h predictions for wells 2–4 are displayed in Figure 11, with the distance to the coast increasing downwards. The time period depicted covers a high-water event and the consequent rise in GWL in the six wells.

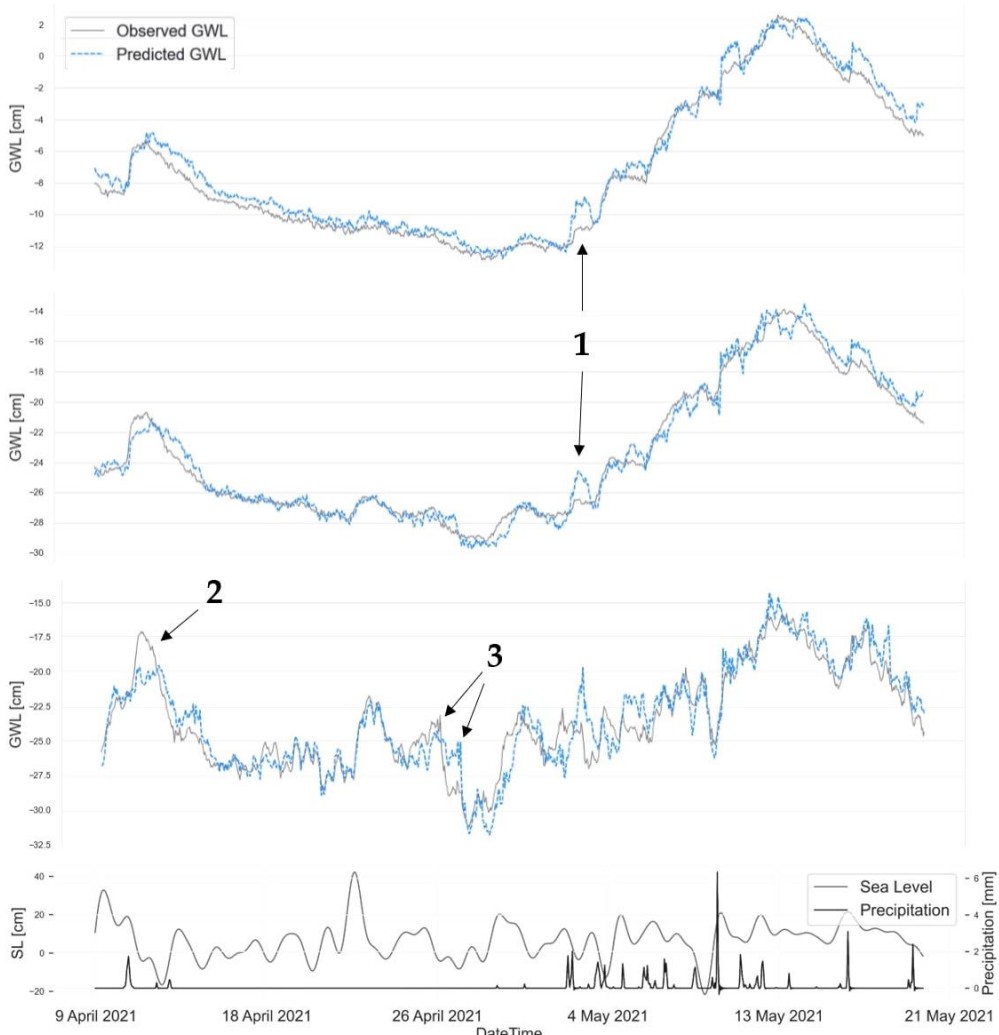

**Figure 11.** The modelled and observed GWL in wells 2 (top), 3 (middle), and 4 (bottom) during SLR and precipitation events. The number 1–3 represent the listed challenges of the model.

The average correlation coefficient for this high-water event was approximately 90%, which indicates the percentage of variations that can be accounted for by the model. The model also performed well, with a high accuracy rate of 78%, in predicting the groundwater level. The average mean deviation was just below 1 cm.

The model, however, has its challenges. Three of the major issues are: (1) An overestimation of the contribution of precipitation; (2) challenges in predicting the extent of extreme GWL rises; (3) relying heavily on the previous groundwater measurements to calibrate the level in the current well. Overestimating the effect of precipitation on the signal can result in periods of a few hours, with significantly higher modelled values than those observed. On the contrary, the inability to model extreme events resulted in modelled values that were lower than the observed values. Both of these factors resulted in a significant deviation between the predicted and measured GWL. The reliability on previous measurements resulted in delayed replications of sudden changes and adjusted the level retroactively.

Table 3 provides a brief summary of the model's overall performance in predicting changes in the groundwater level, specifically for one, three, and five days ahead. The $R^2_{adj}$ values showed that the modelled groundwater level corresponded well with the observed levels, but there were still difficulties with making independent predictions ($R^2_{pred}$). The average deviation from the observed values was only a few centimetres. When attempting to predict groundwater changes three days into the future, the $R^2_{adj}$ decreased, while the ability to make accurate predictions ($R^2_{pred}$) increased. This may be because the model was originally slightly overfitted, resulting in a greater deviation from the observed data but a less-constrained fit of the model when calculating the predicted values.

**Table 3.** The main outcomes of the SVR model predictions of the groundwater level 1, 3 and 5 days ahead using data from logger 4.

| | 1 Day | | 3 Days | | 5 Days | |
| --- | --- | --- | --- | --- | --- | --- |
| | Average | Std. Dev. | Average | Std. Dev. | Average | Std. Dev. |
| $R^2_{adj}$ | 0.76 | 0.19 | 0.43 | 0.37 | −0.27 | 0.78 |
| $R^2_{pred}$ | 0.10 | 0.99 | 0.23 | 0.51 | −0.65 | 1.12 |
| MAE | 1.30 | 0.56 | 3.14 | 1.06 | 4.90 | 1.67 |
| RMSE | 1.99 | 1.07 | 6.94 | 2.45 | 13.02 | 4.29 |

Based on Table 3, it seems unlikely that the model can generate valuable outcomes when trying to forecast groundwater fluctuations five days ahead. Here, both correlation coefficients were negative. However, plotting the model results for the same interval as presented in Figure 11 indicates a different outcome (Figure 12). Despite the greater disparities between the modelled groundwater table and the observed data, the model successfully captured the primary patterns.

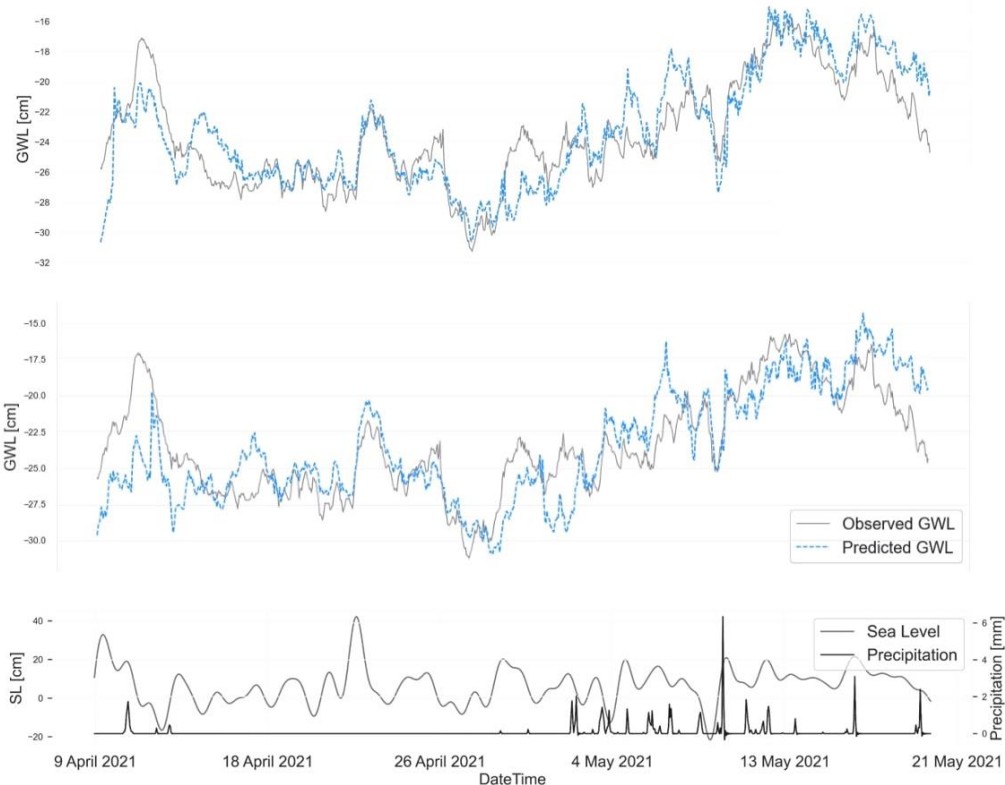

**Figure 12.** The modelled results for 3- and 5-day predictions in logger 4.

The poor performance of the model, as indicated in Table 3, raises the question: Why does it exhibit such bad performance? One possible explanation can be observed in

Figure 13, which depicts a notable and sudden rise in the groundwater level. The model struggled to foresee this abrupt change, resulting in a delayed signal in the predicted groundwater level. As the predictions extended further into the future, the discrepancy between the measured and predicted peaks widened, leading to a decrease in the adjusted correlation coefficient ($R^2_{adj}$). Moreover, the model encountered difficulties in accurately maintaining the average groundwater level over time, causing it to deviate from the true water table. The segment illustrated in Figure 13 serves as an example, among several other instances, that contributes to the reduction of correlation coefficients and the amplification of mean errors. However, this does not imply that the model lacks value; instead, it emphasises the significance of identifying the sources behind any statistical deviations that appear unusual. In this specific case, it may be necessary to investigate whether the sudden increase in the groundwater level is driven by natural factors or human intervention, or whether it requires correction during the data processing stage.

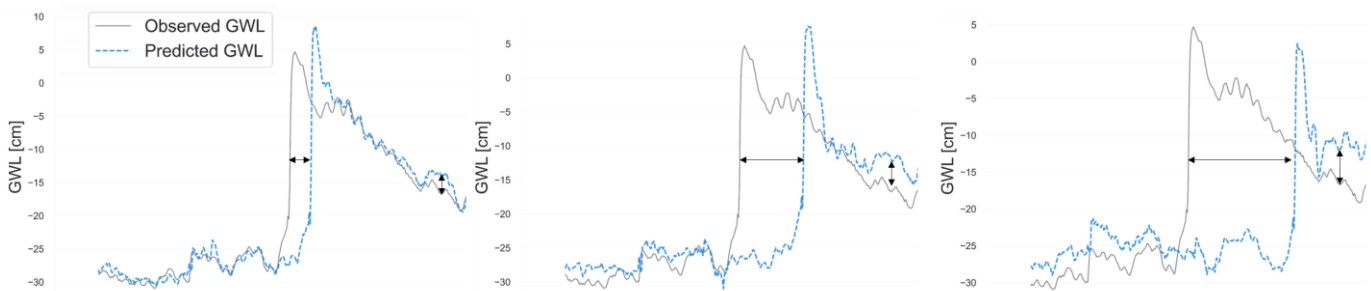

**Figure 13.** The modelled result for 1-, 3- and 5-day predictions during the event of an abrupt increase in the groundwater table. The arrows mark the increasing delay and shift of the model predictions.

## 4. Conclusions

Comparing sea level and groundwater data demonstrated a need for greater emphasis on addressing the flooding caused by groundwater following rises in the sea level—especially in countries with extensive coastlines and low-lying coastal infrastructure. Although the impact of a storm surge is most noticeable in those wells located closest to the coast, responses to the strongest storm surges were additionally registered in those wells furthest away. When there was a clear linear correlation between the delay time and the distance to the coast in an open, undisturbed area, the presence of built constructions appeared to obstruct the signal and cause a further delay.

The response signal experienced a delay that ranges from 14 h and 45 m near the coast to more than 30 h 210 m away from the coast. In several cases, it was difficult to identify the high-water signal in the loggers located far inland, possibly because the signal weakens logarithmically as it moves away from the coast. This causes the impact of factors such as rainfall to become significant enough to have a comparable amplitude, masking the signal from the sea. Meanwhile, a categorisation of the wells suggested that the groundwater level closer to the coast or with a well-established hydraulic connection to the ocean is more heavily influenced by the impact of the sea. Built areas proved to have a significant influence of the signal measured in the groundwater table, either due to low-conductivity barriers or an increased sensitivity to rainwater. In the single wells, there was a semi-linear correlation between the magnitude of the SLR and the corresponding rise in the groundwater table. The events with the smallest rise in sea level caused no or little response in the groundwater wells.

The delayed response allowed for the calculation of a predicted increase in the shallow groundwater subsequent to a sea level rise. The support vector regression model showed promising results when predicting the groundwater response in a logger 24 h in advance using sea level and rainfall data. While the modelled values of the three- and five-day predictions showed little deviation from the observed groundwater table, and single sections of sudden changes in groundwater level resulted in negative correlation coefficients.

It would be interesting to introduce further improvements and move from single-step prediction to predicting entire time sequences. Additionally, expanding the model to a multi-output model calculating the groundwater table in all wells simultaneously would eliminate the need for calibrating each well.

This study highlighted the significance of incorporating the influence of sea level on groundwater variations in the development of predictive models and the examination of future groundwater levels. In certain regions, the interaction between the sea and the shallow groundwater table is intensifying, underscoring the relevance of including this factor in analyses. Furthermore, when determining appropriate climate adaptation strategies, it is crucial to account for the natural dynamics associated with these phenomena.

**Author Contributions:** Conceptualisation, T.R.A., S.E.P. and A.B.M.; methodology, T.R.A., S.E.P., A.B.M. and R.F.M.; formal analysis, R.F.M., H.H.H. and E.P.H.; investigation, R.F.M., H.H.H. and E.P.H.; data curation, R.F.M., H.H.H. and A.B.M.; writing—original draft preparation, R.F.M., E.P.H. and T.R.A.; writing—review and editing, R.F.M., E.P.H., A.B.M. and T.R.A.; visualisation, R.F.M. and T.R.A. All authors have read and agreed to the published version of the manuscript.

**Funding:** This research was funded by EU LIFE (grant number LIFE15 IPC/DK/000006/C2C CC), Insero Horsens (grant number 2020-0044), and Engell Friis Fonden (grant number 2020-021).

**Data Availability Statement:** Parts of the data analysed in this study are publicly available and can be found here: https://www.dmi.dk/kontakt/frie-data/ (accessed on 1 May 2023). The remaining data required to reproduce the findings of this study are available on request from the corresponding author.

**Acknowledgments:** The authors would like to thank Hedensted municipality for their assistance and collaboration during the project. The authors also owe thanks to Kirsten Landkildehus Thomsen and Hans Erik Hansen for carrying out the field and laboratory work.

**Conflicts of Interest:** The authors declare no conflict of interest. The funders had no role in the design of the study; collection, analyses or interpretation of data; writing of the manuscript, or in the decision to publish the results.

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
