# Peer review of "Short-Term Ocean Rise Effects on Shallow Groundwater in Coastal Areas: A Case Study in Juelsminde, Denmark"

_water, doi:10.3390/w15132425_

Round 1

Reviewer 1 Report

This manuscript presents a relevant study on a topic that is not frequently addressed: the effect on groundwater due to storm surges. As the authors introduced, there are important implications for this topic, especially in lowland areas and there is a lack of understanding and methodologies in the research field. The use of machine learning for analyzing the long data series available is a good solution for a first overlook of the processes taking place as well as addressing the predictability of the process. The research is rich in data; it is well-written and presents interesting results that would be useful for the readers of the journal.

I have some minor comments to improve the clarity of the research as well as increase the impact by strengthening the discussion.

General comments

The title can be easily misunderstood; ocean rise is often referring to long-term processes, but what the authors actually researched is the storm surges and even the model is for a 1-5 days period. I think referring to storm surges or being more specific on the time period frame of the study would help the readers to know from the beginning what is the purpose of the study.

I do not see mention of manual measurements for correcting/verifying the series of water levels. I think it would be important to mention it at some point. I assume that it should be some measurements during the period of monitoring.

The discussion in this manuscript is quite reduced, there is no specific section as it is integrated into the results and even if there are some comments along the text that can be considered discussion. It would be beneficial to enlarge this part in the context of international literature to provide a more complete overview of the major outcomes. A few questions to consider are for example (1) how the depth of the boreholes can be distorting the trend for figure 6 regarding distance and response time.  It is not clear the depth of all the boreholes but there are precedents of studies seeing different delays depending on this factor (e.g. Sánchez-Úbeda et al. 2016, Slooten et al., 2010) that can be certainly mentioned as part of the discussion. (2) In figure 7, there are different trends between loggers, some of them seem not to increase at all. Here the topography can play a role. If there is not enough unsaturated zone on top groundwater would not “pile up”. I can see for example that borehole 6 in figure 2 seems to have the water table just at the topographic surface. This has been considered a factor for limiting the increase in groundwater level due to sea level rise in previous studies (Michael et al., 2015; Duque et al, 2022). (3) In the last part of the results section, there are a number of elements showing the uncertainties about the model and the utility for different time periods. This is interesting but it is not presented in any figure other than a table that says that for 5 days the correlation is even negative. It would be relevant to show more specifically these results for the discussion, from a methodological viewpoint it would be interesting to see the different adjustments to the data observation and how it is deviating from the almost perfect matching in the first days. As I recommended deleting a few figures, there is space for adding a new one.

The conclusion section can be improved. Parts of it have not been presented in the manuscript (the delay range), there are some comments very local that cannot be considered a conclusion (one well located 210 m…) or general sentences that are not generated by this research (the greater the magnitude of the sea level, greater the rise in groundwater level). The explanations about the model performance and the outcomes of it can be explained better and the methods and results for differentiating the effect of rain and sea can be resumed in this section.

Specific comments

Line 40. Unclear sentence, say which contributors or delete it.

Lines 58-59. There are also drawbacks on machine learning models as the authors will say later. Machine learning models are not technically substituting the old models so it is not just a replacement one by the other. This sentence can be better explained.

Line 88. Please cite here the small town in Denmark

Lines 106-108. It would be clearer if instead of referring to the Juelsminde area is the study area

Lines 111 and 122. Gyttja is in one case clay and other silt, this can be confusing for the international reader

Figure 2. Add the topography to the figure

Line 151. It is unclear what is to correct to 100 %

Line 155. Therefore, the mean… This sentence is out of context. I do not see the connection between the uncertainty and the subtraction of the mean unless it is said what is the goal of this.

Line 157. What is correct based on the terrain? The terrain does not play any role when talking about groundwater levels.

Line 158. It is confusing that the company that provides the water loggers is also working with the data, then the company is not a provider of loggers, is an integral manager of the data.

Line 218. Before developing a model… This is an obvious sentence, delete.

Line 223. What are the historical data mentioned here? Have they be presented? And real-time data?

Line 224. Precipitation and sea level… this information has been already explained, delete.

Figure 7. Add a legend for the lines presented in the graphs

Line 266. I do not understand how the sea level decreases with the distance to the coast. The sea level is always the same.

Figure 8. Explain or edit graphically to say what is the right graph and the connection with the left one.

Lines 293-296. The blue circles and the triangles are well explained, what about the squares and the white circles?

Lines 303-305. Here the depth can also play a role, see general comments.

Lines 305-307. This is a good explanation on a specific case, adding a few more would give a more complete overview of the different processes taking place in the study area. For example what about well 14? And why 11 and 12 are transitioning so fast to different types and not be a square?

Lines 314-316. All these explanations correspond with a figure caption, they are not needed in the text.

Figures 11 and 12. I do not see it necessary, based on the explanations of the text, to have both figure 11 and 12. They are basically showing the same.

Lines 325-327. I do not see in the figures or in the text, the 3 major issues. The authors can consider marking them in the figures to highlight what are they referring specifically or to explain better the issues. The figures only show a quite good match between the model and the observations.

Lines 335-336. This can be deleted.

Lines 337-349. This belong to methods, not at the end of the results section.

Lines 350. What numbers?

References

Sánchez-Úbeda J.P., Calvache M.L., López-Chicano M., Duque C. 2018. The effect of non-tidal components, the use of peak delays, and depth of measurement for the application of tidal response methods. Water Resources Management 32(2): 481-495.

Slooten L., Carrera J., Castro E., Fernàndez-Garcia D. 2010. A Sensitivity Analysis of Tide-Induced Head Fluctuations in Coastal Aquifers. Journal of Hydrology 393: 370-380.

Michael, H. A., Russoniello, C. J., Byron, L. A. 2013. Global assessment of vulnerability to sea-level rise in topography-limited and recharge-limited coastal groundwater systems, Water Resources Research 49, 22282240.

Duque C., Meyer R., Sonnenborg T.O., 2022. Saltwater intrusion in Denmark. Boletín Geológico y Minero de España 133 (1): 29-46.

Author Response

The file contains replies to both reviewers.

Reviewer 2 Report

The manuscript needs some minor editing before publication. Please, cite the reference for "Juelsminde is prone to flooding and has been identified as one of the top 10 cities in Denmark that are most at risk of flooding" (lines 94-96). Again, cite the references about the geological setting exposed in page 3 (lines 106-123). Background of the Figures 3 and 10 seems to be of low quality resolution. Please, Provide percentages of outliers removed (lines 152-159). Change comma with dot for 1.22m (line 181). In Table 2, add the measurement units for columns 5 and 6. 

Author Response

(The authors gave the same response as above.)

Round 2

Reviewer 1 Report

The authors did an excellent job with the comments. The manuscript has improved greatly and it is a good contribution from the local and international perspective. 

I found a couple of technical errors:

In the figure 13 caption please correct increase: Figure 13. The modelled result for 1-, 3- and 5-day predictions during an event of an abrupt incease 461 of the groundwater table.

Considering the short time for the review it would be good to check the spelling mistakes like this in case there are more. 

In the sentences:

We expect the amplitude damping in a free aquifer to follow a logarithmic function [29]. It think it refers to reference 28

but in

If an aquifer is limited by  the topography, a ground water table close to terrain will leave little room to store more water and hence not have the capacity to let the groundwater table rise in response to the  sea level [30, 31]

I think it refers also to reference 29

and 

All of the events occur in either October, January or early March, which  tend to be cold and wet months in Denmark [32].

I think actually refers to reference 31

Check the references numbers. Actually, with the numbering system of the journal is a good habit to double check that the references numbers are correct. 
